# Analysis of Academic Psychological Variables, Physical Education, and Physical Activity Levels of Mexican Students

**DOI:** 10.3390/bs13030271

**Published:** 2023-03-20

**Authors:** Raúl Baños, Roberto Espinoza-Gutiérrez, Juan José Calleja-Núñez, Gloria Rodríguez-Cifuentes

**Affiliations:** 1Faculty of Sport, Autonomous University of Baja California, Tijuana 22390, Mexico; 2IES Arzobispo Lozano, 30520 Jumilla, Spain

**Keywords:** motivation, basic psychological needs, school satisfaction, self-esteem, emotional intelligence

## Abstract

Physical inactivity is a current worldwide, and especially Mexican adolescents. Therefore, this study has the following objectives: (i) to analyze the PA-LT pattern of Mexican adolescents; (ii) to analyze whether there are differences in emotional intelligence, basic psychological needs, academic motivation, self-esteem, and academic satisfaction according to the PA-LT pattern; (iii) to analyze the relationship between emotional intelligence, basic psychological needs, academic motivation, self-esteem, and academic satisfaction. A total of 748 secondary school students participated, of which 374 were girls (Mean age = 13.99; SD = 0.30) and 374 boys (Mean age = 14.02; SD = 0.33). The questionnaire comprised the following scales: IE, NPB-ESC, EMA, EA, and SIE. The main results obtained show that none of the girls stick to any active physical activity pattern during their spare time. In addition, physically active boys obtained better scores in emotional clarity, emotional repair, perception of academic competence, perception of social relationships, extrinsic motivation, intrinsic motivation, self-esteem, and satisfaction with school.

## 1. Introduction

The teaching–learning process is a key aspect of education. Experts of the National Agency of Education of México focus their attention on the teaching–learning process in order to improve it. However, results obtained from the evaluations at the national level with the National Plan to Assess Learning (PLANEA), and at the international level, from Program for International Student Assessment (PISA), highlight that the teaching–learning process continues to be a worrying aspect of the Mexican secondary education school stage.

In that vein, PLANEA 2017 results reported a decrease in the average results of students compared to the results obtained in 2015. Roughly 47% of students showed a minimum competence in communication and language skills. Mathematics competence is even worse. Just one out of three students gains the skills and competences expected [1]. The average scores obtained throughout PISA carried out for the Organisation for Economic Co-operation and Development (OECD) in its members countries are concerning too. In 2019, Mexican students’ average score showed 420 points in reading literacy, 409 in mathematic literacy and 219 in science literacy. However, the average score of the OECD countries was 487, 489, and 489 [2], respectively. As can be seen, data collected throughout different evaluations reveal some worrying data about the learnings of Mexican adolescents.

The scientific literature has shown the importance of students’ intrinsic satisfaction, development of emotional intelligence, their basic psychological needs in school, academic motivation, self-esteem, and even the physical activity they practice in their leisure time (PA-LT), having an impact on students’ learning process [3,4,5,6,7]. Despite the fact that the international literature is rather clear, there are just a few national researchers that analyze these factors in the Mexican adolescents’ learning process framework [8].

Emotional intelligence is closely related to students′ learning outcomes and is also an all-important aspect of adolescents’ growth [9,10]. According to Salovey et al. [11], emotionally intelligent people have the following traits: (a) ability to accurately perceive, evaluate, and express emotions; (b) ability to flourish emotions that ease thinking processes; (c) ability to understand emotions and self-emotional awareness; and (d) ability to regulate emotions to promote emotional and intellectual growth. It is worth mentioning, that the theoretical emotional intelligence framework proposed by Salovey et al. [11], analyzes the way people think and appreciate their emotions and feelings (emotional attention), the ability a person has to recognize his/her emotions (emotional clarity), and the ability to regulate negative emotional states and widen positive emotions (emotional repair). Given this, emotional intelligence has been related to basic psychological needs and academic motivation [12], self-esteem [13], and adolescents’ satisfaction [14]. Nevertheless, scarce are the studies that analyze, in the school context, emotional intelligence of Mexican adolescents [15].

Regarding motivation, it has been considered one of the factors that best explains human behavior in different life contexts [16], including educational contexts [17]. Self-Determination Theory (SDT) is a macro-theory made up of six mini-theories that studies the motivation of individuals [18,19]. This theory describes that a person can be motivated in three different ways when adopting a behavior: demotivated, extrinsically motivated, and intrinsically motivated [19]. The intrinsic academic motivation supports the idea that the activity is carried out just for self-drive and self-satisfaction, the behavior is adopted for the pleasure experienced during the learning process [20]. In the extrinsic academic motivation, the behavior is adopted based on the external stimuli generated during the activity, that is, based on the rewards or punishments applied to students if they do not achieve what is previously established [21]. Finally, demotivation would reflect the absence of volition and intention to adopt the behavior [19].

Research on the issue has shown that intrinsic motivation decreases during school years, while extrinsic motivation plays a more decisive role, and that motivational swap is more noticeable in the transition stages [22]. On the other hand, academic demotivation is related to a higher dropout rate, lack of attention in class, and resentment towards classmates and teachers [23,24]. It is worth mentioning the importance of students’ feeling motivated, since intrinsic and extrinsic motivation predict the dimensions of the Basic Psychological Need Theory (BPN) [19,25].

The BPN is one of the six mini-theories that make up the SDT. Individuals have an innate tendency toward vitality and effective function to the extent that their BPNs of autonomy, competence, and relatedness are met [18].

Autonomy refers to the willingness of performing any act, whether dependent or independent, collective or individual [19], that is, it refers to the individual’s desire to be the origin of their behavior and, therefore, it is related to the freedom that is granted to make decisions while the activity is carried out [26]. Competence is defined as the individual’s ability to interact effectively with their environment to ensure the conservation of the organism, that is, it provides the energy to learn [18]. Relatedness refers to being connected to and respected by others (for example: classmates and the teacher) and having a feeling of belonging to the group [19].

It has recently been shown that when students feel satisfied with their BPNs in school, this predicts academic self-efficacy and thus improves learning [27]. In addition, it has been proven that BPN satisfaction has a mediating effect between the teacher’s teaching style and intrinsic motivation [28], also relating to self-esteem [29]. However, when students are dissatisfied with BPN, it contributes to psychological distress in adolescents [30].

Regarding self-esteem, the scientific literature has shown the relationship with the previously mentioned variables [19,31]. Melcón and Melcón [32] describe self-esteem as the perception of the characteristics that a person has of his/her own self, revealing inner security, self-confidence, and self-respect. However, at a general level, adolescents experience a poor perception of their abilities, both professionally and personally, creating a negative image of themselves, which leads to having a negative and unambitious future projection of themselves [33].

On the other hand, well-being and academic satisfaction can be analyzed from the Subjective Well-being Theory of Diener and Emmons [34], as a theoretical construct that evaluates the well-being of the adolescent, that is, it analyzes the levels of satisfaction and dissatisfaction of the person, both at a general level with their life, as well as in specific areas of it. Focusing on the satisfaction experienced with school, it has been shown that when Mexican adolescents feel satisfied with their subjects and with the school, they predict academic performance. However, when their well-being at school decreases, this dissatisfaction negatively predicts their academic performance [35], and can even trigger school dropout [36]. In this vein, Baños et al. [37], have recently shown, in a study carried out with Mexican adolescents, that academic and life well-being increase when students are physically active, prompting an opposite effect in more sedentary behaviors.

The scientific literature has shown the great impact that PA-LT has on people’s quality of life. In this way, physical exercise, dance, and participating in physical–creative activities such as games increase levels of emotional intelligence [38], self-esteem [13], satisfaction [37], BPN [39], and motivation, which increases when PA-LT is performed at a higher intensity [40], and increases academic performance, cognitive function, structure, and brain activity in adolescents [41]. However, the PA-LT levels of Mexican adolescents are truly worrisome, since more than 78% claim to be sedentary [37]. In fact, the World Health Organization [42], makes the following physical activity recommendations for adolescents: (a) they should do at least an average of 60 min per day of moderate-to-vigorous intensity, mostly aerobic, throughout the week; (b) they should incorporate vigorous-intensity aerobic activities, as well as those that strengthen muscles and bones, at least 3 days a week; (c) they should limit the amount of time spent being sedentary, particularly the amount of recreational screen time.

Due to the strong relationship between the aforementioned variables, the lack of studies that analyze those variables in the Mexican context, and the high levels of sedentarism in Mexican adolescents, this study is carried out. Due to all these aspects, this study has the following aims: (i) to analyze the PA-LT pattern of Mexican adolescents; (ii) to analyze whether there are differences in emotional intelligence, basic psychological needs, academic motivation, self-esteem, and academic satisfaction based on the PA-LT pattern; (iii) to analyze the relationship between emotional intelligence, basic psychological needs, academic motivation, self-esteem, and academic satisfaction.

## 2. Materials and Methods

### 2.1. Design

The methodology followed a non-experimental, transversal, and correlational–causal design [43]. This investigation was carried out in accordance with the Declaration of Helsinki of 1961 [44].

### 2.2. Participants

The sample was made up of third-year students of a public high school from the State of Nuevo León (Mexico) selected by a probabilistic and multistage design, with stratification at the school level and by proportional allocation. The population framework of students in the third year of secondary school in the State of Nuevo León, according to INEGI sources in 2019, was 27,227 students, where 13,396 (49.2%) were female and 13,831 male. A representative sample indicated according to sex was calculated, for a finite population with a confidence level of 95% and for a margin of error of +5%, being 374 girls with an average age = 13.99 (SD = 0.30) and 374 boys with an average age = 14.02 (SD = 0.33). Selection criteria for recruitment of participants were: (i) being a third-year student of a public high school from the State of Baja California or Nuevo León; (ii) filling out the informed consent, by their parents or guardians, in which the objectives and intention of the study were reflected. An exclusion criterion was that the data collection form had not been duly completed within the different scales. It must be mentioned that 21 individuals did not give their consent to participate in the research and that 34 questionnaires were discarded because they were filled out incorrectly. Lastly, there were no lost values in the responses included in the study.

### 2.3. Instruments

The instruments used were:Trait Meta Mood Scale-24 (EI). To measure emotional intelligence in adolescents, we used the Mexican version of the IE by Valdivia et al. [15]. This instrument presents 24 items that measure the degree of emotional intelligence of adolescents in three dimensions: emotional attention, clarity of feelings, and emotional repair. Each factor is made up of eight items. Responses are collected on a 5-point Likert-type scale ranging from 1 (strongly disagree) to 5 (strongly agree).The BPN scale for school (BPN-SCH) was adapted and validated by Zamarripa et al. [45] to the Mexican school context. This instrument is made up of three subscales that measure the needs for autonomy, competence, and social relationships throughout a total of 16 items with a 7-point Likert-type response scale, ranging from 1 (strongly disagree) to 7 (strongly agree).Academic Motivation Scale (AMS). To measure the student’s academic motivation, the Spanish version translated and validated for the secondary education stage was used [21]. The AMS is made up of 28 items that measure intrinsic motivation, extrinsic motivation, and demotivation. Responses were scored on a 7-point Likert-type scale, from 1 (“Does not correspond at all”) to 7 (“Completely corresponds”).Self-esteem scale (SE). To measure self-esteem, we used the Spanish version of Atienza et al. [46]. This instrument presents 10 items, 5 written positively and 5 negatively, which measure the degree of global self-esteem. Responses are collected on a 4-point Likert-type scale ranging from 1 (strongly disagree) to 5 (strongly agree).Intrinsic Satisfaction Classroom Scale (ISC). To measure intrinsic satisfaction with the school, the ISC instrument was used, validated for the Mexican context by Baños et al. [35]. This scale is made up of eight items, of which five integrate the factor referring to the level of satisfaction/fun with the school, and three the factors referring to dissatisfaction/boredom. Responses are collected using a scale ranging from 1 (strongly disagree) to 5 (strongly agree).Levels of physical activity during leisure time. To know the levels of physical activity of the students, five questions are used in an index of amount of physical activity [47]. The items measure frequency, duration, and intensity of physical exercise during leisure time and participation in organized sports and sports competitions. The highest results indicate that the individuals are physically active, and the lowest that they are more sedentary.

### 2.4. Procedure

This research was conducted in accordance with the 1961 Declaration of Helsinki [44]. The approval for this study was obtained by the Mexican Ministry of Public Education and the Autonomous University of Baja California (identification number: 431/569/E). To carry out this research, a research project was presented to the Secretary of Public Education of Mexico called: “Program for International Student Assessment: relationship between school performance in secondary school students with psychological, family and physical activity variables” which was approved and subsidized by the aforementioned body.

Subsequently, permission was requested from the heads of the secondary education schools, giving the parents or guardians of the students an informed consent form in which the objectives and intention of the study were reflected. Once the aforementioned permits were obtained, the data were collected, informing the participants. The participation was anonymous and voluntary and the treatment of their answers was confidential, informing them that there were no correct or incorrect answers, and requesting that they answer with the utmost sincerity. The questionnaires were completed in the classroom, with the principal investigator always present to answer any question during the process, lasting 15–20 min.

### 2.5. Data Analysis

Regarding the statistical analyses conducted, first, frequency and descriptive analyses of the PA-LT Levels scale were performed, studying whether there were significant differences based on gender through the χ^2^ statistic. Subsequently, the descriptors of each subscale (IE, NPB-ESC, AMS, EA, and SIE) were analyzed, studying the indices of skewness, kurtosis, and the Kolmogorov–Smirnov test to study the normality of the data and internal consistency using the Cronbach test’s alpha for the study of trust. After the results obtained from the normality test, the Mann–Whitney U test was applied to contrast the measurements in pairs, taking each of the factors in the questionnaire as dependent variables and considering the pattern of physical activity in leisure time as a grouping variable, with Spearman correlation analysis to study the relationship between the variables. For all the analyses, the SPSS v.25 package was used.

## 3. Results

Firstly, the results obtained on the pattern of behaviors in the practice of PA-LT are described, which indicate significant differences (*p* < 0.001) depending on gender (see Table 1). Both girls and boys present a rather worrying picture, especially girls, with 100% who declare themselves physically inactive. Furthermore, 70% of the boys declare themselves physically inactive in their leisure time.

In Table 2, we can see the mean values, standard deviation, skewness indices, internal consistency, and the Kolmogorov–Smirnov normality test for each instrument factor. Regarding internal consistency, almost all Cronbach’s alpha (α) values are above the acceptable values according to Dunn et al. [48] and Hair et al. [49], although the boredom dimension of the SIE scale obtained values below 0.70. Skewness indices should be close to 0 below 2. The results of the Kolmogorov–Smirnov test indicated that the data presented a non-normal distribution, and for that reason, the Spearman coefficient was used to study the correlations.

To analyze the differences between the dimensions based on the PA-LT pattern of the students, the Mann–Whitney U statistical test was used, as shown in Table 3. The PA-LT pattern of the adolescents is shown as a determining factor to obtain significant differences. The students who reported being physically active obtained higher scores in the dimensions of clarity and emotional repair, perception of competence in school, extrinsic and intrinsic motivation, and satisfaction with school than those students who reported being physically inactive.

The results obtained from the correlation analysis can be seen in Table 4.

## 4. Discussion

The main results of this research show the worrying levels of physical inactivity in female Mexican adolescents, since none self-reported meeting the minimum levels of daily physical activity. In addition, physically active boys obtained better scores in emotional clarity, emotional repair, perception of academic competence, perception of social relationships, extrinsic motivation, intrinsic motivation, self-esteem, and satisfaction with school.

Regarding the first aim, this research shows an alarming situation, since 100% of the girls and more than 70% of the boys declare themselves as physically inactive. Similar results are shown in other studies also carried out with Mexican adolescents in which girls had higher levels of physical inactivity than boys [37,50,51]. A possible explanation for these results could be that Mexican girls experience less satisfaction and fun in PE lessons than boys, so Mexican girls feel more boredom in the PE lessons [37,52]. In this vein, Ntoumanis et al. [53], highlight the importance that adolescents have fun during the PE lessons, so that PA-LT increases. In addition, concern grows over the sedentary behavior of Mexican students because they have no future intention to practice PA-LT [54]. A possible explanation for these sedentary lifestyles of Mexican adolescents could be found in the study carried out by Baños [55]. This author compared physical education lessons in Mexico with those in Spain and found that Mexican physical education teachers design lessons prioritizing the comparison of skills among students, and Spanish teachers focus on designing lessons where the priority is that students perceive that they are self-improving their skills, comparing themselves with their own level and not with the rest of their classmates. In this line, when the comparison between students is prioritized, frustration is generated in those who are less competent at a motor level, and this is related to a higher drop-out rate of the PA-LT in Mexican adolescents [56,57].

Concerning the second aim, the data obtained in the present study showed that the PA-LT pattern of adolescents is shown to be a determining factor, generating significant differences. The students who reported being physically active obtained higher scores in the dimensions of emotional clarity and repair, perception of competence in school, extrinsic and intrinsic motivation, and satisfaction with the school than those students who reported being physically inactive. Similar results were found by Galdón et al. [58] in a study with Spanish adolescents, relating the dimensions of emotional intelligence to a greater pattern of PA-LT and a greater intensity of it. In addition, if these physical–recreational activities are of a creative nature such as dance or games, this also affects levels of emotional intelligence [38]. Regarding basic psychological needs, studies have shown that, when students have a higher rate of PA-LT practice, they increase the satisfaction of basic psychological needs in physical education [39,57,59]. Similar results were obtained by Vazou et al. [60], in which academic motivation improved after implementing an integrated physical activity program in mathematics, social science, and language lessons. However, the present study did not find a relationship between self-esteem and PA-LT, a relationship that other studies found [13]. Likewise, Baños et al. [37], obtained similar results, in that students who showed a higher PA-LT rate felt more satisfied with school. In this way, the results obtained in the present investigation provide relevant information for the Mexican educational system, since they show that secondary school students with a higher rate of PA-LT also have higher levels of emotional intelligence, satisfaction of the BPN, academic motivation, and school satisfaction.

In relation to the third aim of this study, it is verified that all the dimensions of the EI scale are positively and statistically significantly correlated with all the dimensions of the BPN scale, with the intrinsic and extrinsic motivation dimensions of the AM scale and with the school satisfaction of the ISC scale. Similar results were obtained by other studies, where emotional intelligence was correlated with basic psychological needs and academic motivation [12] and adolescent satisfaction [14]. This relationship could be due to the different facts such as when students develop greater control of their emotions, their ability to make decisions improves, they are able to focus their attention better, they are also more capable of focusing on their skills to make decisions without being emotionally affected by their lack of competence, and they can develop greater social skills, and therefore, their academic motivation increases [61,62].

On the other hand, only the dimensions of clarity and emotional repair are positively and statistically significantly correlated with students’ self-esteem, finding no relationship with the dimension of emotional attention. In this line, Umstattd et al. [13] found a relationship with all the dimensions of emotional intelligence, so it is necessary to continue conducting studies that relate the variable of emotional attention with self-esteem. A possible explanation for the data obtained in this research could be that, when the person clearly identifies their feelings and is able to focus on positive emotions reversing the negative emotions, self-confidence increases and so does self-esteem.

With regard to the demotivation dimension of the AMS scale, it was negatively and statistically significantly correlated with emotional repair, perception and relationship of competence, extrinsic and intrinsic motivation, self-esteem, and satisfaction with school, and positively and statistically significantly with boredom at school. On the contrary, the dimensions of intrinsic and extrinsic motivation and satisfaction with the school correlated positively and statistically significantly. Similar results were obtained from other studies [19,20,23,24,25,63]. This could be because, when students feel satisfied at school, they enjoy learning, so their motivation and self-esteem increases. On the contrary, if an adolescent feels bored, due to monotonous and repetitive lessons, their motivation will decrease.

Finally, it is worth mentioning that boredom at school correlated negatively and statistically significantly with the perception of autonomy and relationships, intrinsic and extrinsic motivation, and satisfaction with the school. Other studies obtained similar results [19,20,25,63]. This relationship could find an explanation on the teacher’s side, that is, when the teacher is authoritarian, this does not allow students to make decisions and, in addition, there is not a dynamic and proactive climate in the classroom, so student motivation decreases and boredom increases at school.

This research has a set of limitations that should be mentioned. The data obtained from this research were collected from third-year high school students in the State of Nuevo León, so they cannot be generalized to the other States of the Federal Republic of Mexico. In addition, it has not been possible to analyze the differences in the variables between active and inactive girls, since they all reported being physically inactive. However, the sample design stands out, being probabilistic and random by centers, stratified, multistage, and by proportional allocation. In that sense, the results of this study can be generalized to the State of Nuevo León, in Mexico. Moreover, another important strength of this study is the topic it addresses, since it can greatly contribute to providing solutions to some problems related to the learning process of Mexican adolescents at school.

## 5. Conclusions

In conclusion, the results of this study provide great information about the PA-LT pattern of Mexican adolescents and the psychological variables that analyze their feelings within the school. The main results obtained in the research highlight that all the girls surveyed stated that they were physically inactive. Students who were physically active obtained higher scores on the dimensions of emotional clarity and repair, perception of competence in school, perceived social relationships, self-esteem, extrinsic and intrinsic motivation, and satisfaction in school than those students who reported being physically inactive. However, it is worrying the percentage of Mexican adolescents who describe themselves as having sedentary behaviors. This higher rate does not only have a bad impact on their psychological and physiological health, but also on their motivation and academic satisfaction. Finally, it is worth mentioning the relationship found between emotional intelligence, basic psychological needs, motivation, self-esteem, and satisfaction with school, since it affects the academic daily life of adolescents.

## Figures and Tables

**Table 1 behavsci-13-00271-t001:** Chi-square (χ^2^) by sex of the pattern of physical activity during leisure time.

	Girls		Boys		χ^2^	*p*
	*n*	%	*n*	%		
Pattern of physical activity during leisure time						
Inactives	374	100%	265	70%	131,896	0.000
Actives	0	0%	113	30%		

**Table 2 behavsci-13-00271-t002:** Descriptive analysis, internal consistency, and normality of the EI, BPN-SCH, AMS, SE, and ISC scales.

Scales	Dimensions	M	SD	α	S	K	Z
EI	Emotional attention	3.48	0.78	0.81	−0.35	−0.28	0.000
	Emotional clarity	3.57	0.80	0.83	−0.46	−0.15	0.000
	Emotional repair	3.77	0.80	0.82	−0.62	−0.28	0.000
BPN-SCH	Autonomy perception	3.69	0.77	0.74	−0.58	0.15	0.000
	Competence perception	3.94	0.67	0.76	−0.76	0.81	0.000
	Relationship perception	3.99	0.73	0.70	−0.88	0.86	0.000
AMS	Demotivation	2.38	1.67	0.85	0.98	−0.36	0.000
	Extrinsic motivation	5.65	0.96	0.87	−0.89	0.71	0.000
	Intrinsic motivation	5.20	1.06	0.89	−0.56	−0.05	0.000
SE	Self-esteem	3.06	0.51	0.74	−0.38	−0.12	0.000
ISC	Academic satisfaction	3.45	0.86	0.77	−0.26	−0.15	0.000
	Academic boredom	2.94	1.04	0.68	0.03	−0.70	0.000

*Note:* M = Mean; SD = Standard Deviation; α = alfa de Cronbach; S = Skewness; K = Kurtosis; Z = Kolmogorov–Smirnov.

**Table 3 behavsci-13-00271-t003:** Differences depending on the levels of physical activity during leisure time.

Scales	Dimensions	Inactives		Actives		U	Z	*p*
		M	SD	M	SD			
EI	Emotional attention	3.32	0.79	3.42	0.75	14,186.5	−0.81	0.418
	Emotional clarity	3.51	0.86	3.77	0.67	12,538.0	−2.51	0.012 *
	Emotional repair	3.64	0.84	3.94	0.72	11,627.5	−3.44	0.000 ***
BPN-SCH	Perception of autonomy	3.63	0.80	3.79	0.72	13,114.0	−1.91	0.055
	Perception of competence	3.90	0.72	4.07	0.58	12,938.5	−2.10	0.036 *
	Relationship perception	3.81	0.77	4.02	0.67	12,702.0	−2.34	0.019 *
AMS	Demotivation	2.79	1.76	2.55	1.75	14,016.0	−1.00	0.317
	Extrinsic motivation	5.54	0.91	5.81	0.93	12,069.0	−2.99	0.003 **
	Intrinsic motivation	5.03	1.07	5.46	1.00	11,144.0	−3.939	0.000 ***
SE	Self-esteem	3.02	0.54	3.15	0.43	12,837.0	−1.99	0.046 *
ISC	Academic satisfaction	3.27	0.91	3.73	0.77	10,558.0	−4.55	0.000 ***
	Academic boredom	3.07	1.07	2.95	0.96	13,825.5	−1.19	0.236

*Note:* M = Mean; SD = Standard Deviation; U = U de Mann–Whitney; Z = Reason; * (*p* < 0.05), ** (*p* < 0.01), *** (*p* < 0.001).

**Table 4 behavsci-13-00271-t004:** Correlation between the dimensions of the scales.

	EC	ER	A	C	R	DES	EM	IM	SE	AS	AB
EA	0.390 **	0.329 **	0.311 **	0.279 **	0.276 **	0.051	0.265 **	0.280 **	0.025	0.216 **	0.065
EC		0.524 **	0.490 **	0.448 **	0.393 **	−0.066	0.245 **	0.293 **	0.351 **	0.331 **	−0.084
ER			0.415 **	0.455 **	0.450 **	−0.109 **	0.275 **	0.346 **	0.397 **	0.379 **	−0.066
A				0.633 **	0.541 **	−0.063	0.439 **	0.487 **	0.360 **	0.480 **	−0.156 **
C					0.532 **	−0.162 **	0.414 **	0.464 **	0.454 **	0.465 **	−0.058
R						−0.189 **	0.376 **	0.447 **	0.427 **	0.362 **	−0.201 **
DES							−0.175 **	−0.127 **	−0.353 **	−0.128 **	0.310 **
EM								0.709 **	0.269 **	0.356 **	−0.141 **
IM									0.248 **	0.524 **	−0.246 **
SE										0.281 **	−0.273 **
AS											−0.178 **
AB											

*Note:* ** (*p* < 0.01); EA = Emotional attention; EC = Emotional clarity; ER = Emotional repair; A = Autonomy; C = Competence; R = Relatedness; DES = Demotivation; EM = Extrinsic motivation; IM = Intrinsic motivation; SE = Self-Esteem; AS = Academic satisfaction; AB = Academic boredom.

## Data Availability

The data presented in this study are available on request from the corresponding author. The data are not publicly available due to privacy.

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
