# Peer review of "Analysis of Academic Psychological Variables, Physical Education, and Physical Activity Levels of Mexican Students"

_behavsci, 2023, doi:10.3390/bs13030271_

Round 1

Reviewer 1 Report

I would like to recognize the authors efforts, since this there is a lack of research about this topic in Mexico. The theoretical framework is congruent with the topic, the research design, results and conclusion are consistent. Along with the results and considering the study limitations, I would like to encourage the authors to keep this line of research for further studies. There is only a minior suggestion

Author Response

We would like to acknowledge the reviewer's comment.
We are aware of the great problem of sedentary lifestyle in Mexican society and our intention is to continue with this line of research.

Reviewer 2 Report

Dear authors, you address a very important topic in your manuscript. In this form, the article is not recommended by me for printing. Revisions should be made in accordance with my suggestions:

Materials and methods:

-please add in the paper in this section a graph showing the selection of the research sample and the course of the experiment

- what inclusion and exclusion criteria were used in the study (please put them on the graph as well). at no place in the paper are these described 

- what steps were taken to minimize measurement error during the implementation of the experiment?

what was the % return of completed questionnaires. How many questionnaires were rejected from the analysis and why?-please add in this section tables of characteristics of the research group including gender, age place of residence and other sociometric data

Results

please present the most important results in the form of graphs. In addition, in the tables please include the studied parameters by gender of the respondents. Are there significant correlations and differences between boys and girls in terms of study variables?

these results are missing in this section, which would significantly increase the scientific value of this work

Discussion

- Please elaborate on the reference to the variables studied in relation to gender in the discussion. The authors only mention the differentiation on gender in relation to physical activity.What about the other variables and comparing them according to the gender of the subjects.Both girls and boys were studied.

Are there any correlations in girls and boys?

Limitations

In parcy the authors did not pdo any limitations.Please add such a section and describe it in detail

Author Response

We would like to acknowledge the suggestions made by the reviewer, as they considerably improve the manuscript.

Point 1: Please add in the paper in this section a graph showing the selection of the research sample and the course of the experiment.

Response 1: Please provide your response for Point 1. (in red)

Point 2: What inclusión and exclusion criteria were used in the study (please put them on the graph as well). at no place in the paper are these described

Response 2: The following paragraph was introduced in the participants section:

"Selection criteria for recruitment of participants were: (i) being a third-year student of a public high school from the State of Baja California or Nuevo León; (ii) filling out the informed consent, by their parents or guardians, in which the objectives and intention of the study were reflected. An exclusion criterion was that the data collection´s form had not been duly completed within the different scales".

Point 3: What steps were taken to minimize measurement error during the implementation of the experiment?

Response 3: In the procedure section it is specified that the main investigator was always present during the data collection to resolve any participants´doubt or question, they may have, regarding the self-report questionnaire.

Point 4: What was the % return of completed questionnaires. How many questionnaires were rejected from the analysis and why?-please add in this section tables of characteristics of the research group including gender, age place of residence and other sociometric data

Response 4: The following information was included in the participants´ section:

It must be mentioned that 21 individuals did not give their consent to participate in the research and that 34 questionnaires were discarded because they were filled out incorrectly. Lastly, there were no lost values in the responses included in the study.

Point 5: Please present the most important results in the form of graphs. In addition, in the tables please include the studied parameters by gender of the respondents. Are there significant correlations and differences between boys and girls in terms of study variables? these results are missing in this section, which would significantly increase the scientific value of this work

Response 5: In line 304, we have added a brief discussion based on gender.

We appreciate the reviewer's suggestion and consider it appropriate, in principle. However, after checking and contrasting the data with the graphics, they are not really eye-catchy. This is because none of the girls were physically active.

We consider that the questions posed by the reviewer are successful and interesting. However, the objectives of the present study are:

  • to analyze the PA-LT pattern of Mexican adolescents;
  • to analyze whether there are differences in emotional intelligence, basic psychological needs, academic motivation, self-esteem, and academic satisfaction based on the PA-LT pattern;
  • to analyze the relationship between emotional intelligence, basic psychological needs, academic motivation, self-esteem, and academic satisfaction

We do not refer to differences depending on sex, but to physical activity patterns during free time (physically active / inactive). If the reviewer considers that we must add new research and new analysis objectives, we are willing to do it. However, we consider that we would be deviating from the issue of the manuscript, having as a final result, a manuscript too extensive.

Point 6: Please elaborate on the reference to the variables studied in relation to gender in the discussion. The authors only mention the differentiation on gender in relation to physical activity.What about the other variables and comparing them according to the gender of the subjects.Both girls and boys were studied. Are there any correlations in girls and boys?

Response 6: We consider that this point is related to the previous answer. In any case, if the reviewer considers that we must make changes in the objectives of the research and in the analysis we are willing to carry them out.

Point 7: Limitations. In parcy the authors did not pdo any limitations.Please add such a section and describe it in detail

Response 7: A study limitations´ paragraph has been added at the end of the discusión

Reviewer 3 Report

The manuscript is interesting but I have several concerns regarding the data analysis for boys vs. girls. Please, see all my comments below:

Abstract

The abstract should be clear and concise and not merely a copy and paste from other sections of the manuscript

Introduction

The introduction is too long and out of focus

Use the WHO guide for describing physical activity in adolescents. Also describe (according to WHO guidelines) appropriate levels of intensity and duration of physical activity, in addition to strengthening the benefits of physical activity

If possible, define and explain the difference between physical exercise, dance, and physical-creative activities (recreational activities?)

Please state the study hypothesis (if applicable)

Methods

Is the selected school public/private? Does physical activity levels differ among public/private school students?

Are there other socioeconomic/cultural factors that may influence Mexican adolescents' low adherence to physical activity in addition to the aforementioned methodological difference between Mexico and Spain?

Results

Results from Table 2 could be under the methods section since they do not add any significant information to the results

It is unclear whether the results in Table 3 are derived from both boys and girls. This information should be evident in the text since 100% of the girls are inactive (and this comparison makes no sense for girls). This comparison should be made only for boys.

Please indicate the abbreviations under Table 3 legend

Lines: 245 to 258: is it necessary to repeat all the information already indicted in Table 4?

Table 4: replace commas for dots

Discussion

The first paragraph is repetitive; please, replace it with an overview of the main findings of the study

Paragraph starting at line 287 should discuss data from boys and girls individually

Please discuss articles that bring the difference between physically active x inactive girls as compared to physically active vs inactive boys

Conclusion

It should be noted that all the female respondents claimed to be physically inactive, so there is no way to know the response pattern of physically active girls and whether there is a difference between this pattern when compared to boys

Perhaps the limitations of the study should appear under the discussion section (not under the conclusion section)

Author Response

We would like to acknowledge the suggestions made by the reviewer, as they considerably improve the manuscript.

Point 1: The abstract should be clear and concise and not merely a copy and paste from other sections of the manuscript.

Response 1: The abstract has been modified

Point 2: The introduction is too long and out of focus

Response 2: Please provide your response for Point 2. (in red)

Point 3: Use the WHO guide for describing physical activity in adolescents. Also describe (according to WHO guidelines) appropriate levels of intensity and duration of physical activity, in addition to strengthening the benefits of physical activity

Response 3: The reviewer's suggestions were made

Point 4: If posible, define and explain the difference between physical exercise, dance, and physical-creative activities (recreational activities?)

Response 4: This is the only point suggested by the reviewer that we do not consider important to add to the introduction. In this study, any type of physical activity that adolescents have done in their free time has been measured, without making differences between physical exercise, dance, physical-recreational activities. Participants were simply asked if they did any type of physical activity outside of school hours, without differentiating what type of activities they did. In any case, if the reviewer considers it appropriate to describe the differences, we will do it, in the next round of review.

Point 5: Is the selected school public/private? Does physical activity levels differ among public/private school students?

Response 5: The selected schools were public. We do not know if there are differences between public and private institutions

Point 6: Are there other socioeconomic/cultural factors that may influence Mexican adolescents' low adherence to physical activity in addition to the aforementioned methodological difference between Mexico and Spain?

Response 6: This is a good question. The reviewed scientific literature states that Spanish students are less bored and have more fun in PE lessons, unlike Mexicans. It should also be noted that Spanish students perceive that the learning climate in PE classes is more focused on mastery. However, Mexican students perceive the learning climates in PE as more focused on performance.

In addition, we believe that aspects such as safety on the streets can influence, since Mexican adolescents do not have as much freedom to go out as the Spanish ones. Insecurity is a big issue in the country. This is a hypothesis of the authors, we cannot confirm it.

In fact, we hope to continue this line of research to try to answer these questions that are still unknown.

Point 7: Results from Table 2 could be under the methods section since they do not add any significant information to the results.

Response 7: After reviewing this point in different manuscripts from the Behavioral Sciences journal, we consider it appropriate to keep Table 2 in the results section, since other manuscripts published in the same journal also have it in that section.

Point 8: It is nuclear whether the results in Table 3 are derived from both boys and girls. This information should be evident in the text since 100% of the girls are inactive (and this comparison makes no sense for girls). This comparison should be made only for boys.

Response 8: We thank the reviewer for the observation as we had not noticed this error. Table 3 has been modified taking into account only the sample of children

Point 9: Please indicate the abbreviations under Table 3 legend

Response 9: The abbreviations were indicated

Point 10: Lines: 245 to 258: is it necessary to repeat all the information already indicted in Table 4?

Response 10: Modified: The results obtained from the correlation analysis can be seen in Table 4.

Point 11: Table 4: replace commas for dots

Response 11: The suggestions have been taken into account.

Point 12: The first paragraph is repetitive; replace it with an overview of the main study findings

Response 12: The reviewer's suggestions were made

Point 13: Paragraph starting at line 287 should discuss data from boys and girls individually

Response 13: The reviewer has made a very successful observation since we had not discussed the results of the first objective depending on sex. However, we have not done it in the paragraph where the reviewer advised us to carry it out , since this paragraph refers to objective 2: (ii) to analyze whether there are differences in emotional intelligence, basic psychological needs, academic motivation, self-esteem and academis satisfaction according to the PA-LT pattern.

As we can see that paragraph refers to the differences depending on whether adolescents are physically active or inactive.

Therefore, we have taken into consideration, this suggestion of the reviewer in the previous paragraph

Point 14: Please discuss articles that bring the difference between physically active x inactive girls as compared to physically active vs inactive boys

Response 14: In line 304, we have added a brief discussion based on gender.

We appreciate the reviewer's suggestion and consider it appropriate, in principle. However, after checking and contrasting the data with the graphics, they are not really eye-catchy. This is because none of the girls were physically active.

We consider that the questions posed by the reviewer are successful and interesting. However, the objectives of the present study are:

  • to analyze the PA-LT pattern of Mexican adolescents;
  • to analyze whether there are differences in emotional intelligence, basic psychological needs, academic motivation, self-esteem, and academic satisfaction based on the PA-LT pattern;
  • to analyze the relationship between emotional intelligence, basic psychological needs, academic motivation, self-esteem, and academic satisfaction

We do not refer to differences depending on sex, but to physical activity patterns during free time (physically active / inactive). If the reviewer considers that we must add new research and new analysis objectives, we are willing to do it. However, we consider that we would be deviating from the issue of the manuscript, having as a final result, a manuscript too extensive.

Point 15: It should be noted that all the female respondents claimed to be physically inactive, so there is no way to know the response pattern of physically active girls and whether there is a difference between this pattern when compared to boys

Response 15: The reviewer's suggestions were made

Point 16: Perhaps the limitations of the study should appear under the discussion section (not under the conclusión section)

Response 16: The reviewer's suggestions were made

Round 2

Reviewer 2 Report

Dear Authors,

Thank you for addressing my comments and making corrections,

the manuscript has gained in content and is publishable

in future studies, please take into account my suggestions to expand them and show intergenerational differences

Author Response

We want to thank the work done by the reviewer. He has substantially improved the manuscript. This manuscript is part of a macro project. In addition, in the future, we will continue with this line of research and the observations made by the reviewer, we will take them into account since we consider them important.

Reviewer 3 Report

The authors accepted some of my suggestions. I still think that the comparison of boys vs. girl should be explored. Best of luck in future studies.

Minor 

It looks like Table 1 and Table 3 were not indicated in the text

Author Response

We want to thank the work done by the reviewer. He has substantially improved the manuscript. This manuscript is part of a macro project. In addition, in the future, we will continue with this line of research and the observations made by the reviewer, we will take them into account since we consider them important.

"Table 1" "Table 3" has been entered in the text